# The Role of Circulating Tumor Cells in the Prognosis of Metastatic Triple-Negative Breast Cancers: A Systematic Review of the Literature

**DOI:** 10.3390/biomedicines10040769

**Published:** 2022-03-25

**Authors:** Lorena Alexandra Lisencu, Sebastian Trancă, Eduard-Alexandru Bonci, Andrei Pașca, Carina Mihu, Alexandru Irimie, Oana Tudoran, Ovidiu Balacescu, Ioan Cosmin Lisencu

**Affiliations:** 1Department of Oncological Surgery and Gynecological Oncology, “Iuliu Hațieganu” University of Medicine and Pharmacy, 400012 Cluj-Napoca, Romania; lisencu.lorena@umfcluj.ro (L.A.L.); pasca_andrei@elearn.umfcluj.ro (A.P.); airimie@umfcluj.ro (A.I.); ioan.lisencu@umfcluj.ro (I.C.L.); 2Department of Anaesthesia and Intensive Care II, “Iuliu Hațieganu” University of Medicine and Pharmacy, 400012 Cluj-Napoca, Romania; 3Department of Surgical Oncology, The Oncology Institute “Prof. Dr. Ion Chiricuță”, 400015 Cluj-Napoca, Romania; 4Department of Toxicology and Clinical Pharmacology, “Iuliu Hatieganu” University of Medicine and Pharmacy, 400012 Cluj-Napoca, Romania; mihu.carina@elearn.umfcluj.ro; 5Department of Genetics, Genomics and Experimental Pathology, The Oncology Institute “Prof. Dr. Ion Chiricuță”, 400015 Cluj-Napoca, Romania; oana.tudoran@iocn.ro (O.T.); ovidiubalacescu@iocn.ro (O.B.); 6Department of Medical Oncology, “Iuliu Hațieganu” University of Medicine and Pharmacy, 400012 Cluj-Napoca, Romania

**Keywords:** metastatic TNBC, CTCs, progression-free survival, overall survival, therapy response

## Abstract

Breast cancer is one of the leading causes of death in women worldwide. One subtype of breast cancer is the triple-negative, which accounts for 15% of total breast cancer cases and is known for its poor prognosis. The main cause of death is due to metastasis. Circulating tumor cells (CTCs) play a key role in the metastatic process. CTCs arise either by detaching from the primary tumor or from cancer stem cells undergoing an epithelial-to-mesenchymal transition (EMT). This review aims to present up-to-date data concerning the role of CTC numbers in relation to the prognostic and treatment response in metastatic triple-negative breast cancer (mTNBC) patients, and also to discuss the methods used for CTCs’ identification. A search in the MEDLINE database was performed. A total of 234 articles were identified. The results of the 24 eligible studies showed that positive CTC status is associated with shorter overall survival (OS) and progression-free survival (PFS) in mTNBC patients. Furthermore, a decrease in number of CTCs during therapy seems to be a favorable prognostic factor, making CTCs’ detection an important prognostic tool before and during therapy in mTNBC patients. The methods used for CTC detection are still developing and need further improvement.

## 1. Introduction

Breast cancer, one of the leading causes of death in women worldwide [1], is a highly heterogeneous disease, with specific histological, molecular, and clinical features [1,2,3,4]. Although previous studies pointed out new insights concerning the molecular classification of breast cancer, therapy is still based on the expression of hormone receptors (HR), estrogen receptors (ER), and progesterone receptors (PR), as well as the human epidermal growth receptor (HER2) [5]. Furthermore, another decisional factor for treatment choice is breast cancer staging. Staging shows the extent of the disease, and the most frequently used system is TNM in which T stands for tumor size, N for the spread of lymph nodes, and M for distant metastasis. Five stages of breast cancer are described: 0 (in situ carcinoma), I and II (early stage invasive), III (locally advanced), and IV (metastatic cancer). The prognosis of the disease worsens with progression to higher stages, with a five-year survival of almost 100% in stage 0 to 23.4% in stage 4 [6]. However, because the treatment response drastically differs between breast cancer subtypes, new molecular features have to be identified to characterize and treat each subtype better [7]. As a result, a comprehensive analysis of The Cancer Genome Atlas (TCGA) data, incorporating information about genetics, epigenetics, transcriptomics, and protein array, was previously conducted [8]. Although new molecular data were included in each molecular subtype, no significant progress was made concerning targeted therapies. As a result, the TNBC subtype continues to present poor clinical outcomes and an increased risk of metastatic spread [9]. The TNBC subtype accounts for 15% of breast cancer cases [7]. The main cause of death in breast cancer is represented by metastasis [10].

A group of cells with the ability of tumorigenesis and metastatic potential is described in breast cancer as cancer stem cells (CSCs) [10,11,12]. These cells hold the ability of self-renewal and tumorigenesis [10]. A part of these cells undergo an epithelial–mesenchymal transition (EMT) and become circulating tumor cells (CTCs) [13,14,15,16]. During EMT, CSCs undergo changes that lead to a shift from a stationary state into a migratory one. In this way, they will migrate through the bloodstream to distant sites where the metastatic process will start. Although the mechanism of the EMT is not completely understood, a downregulation of E-cadherin takes place. Therefore, cells detach from the primary tumor, lose their epithelial features, and gain mesenchymal ones. When tumor cells reach the second sites, the reverse process of mesenchymal-epithelial transition (MET) takes place; cells are regaining the epithelial features and start the new tumor growth [17]. CTCs are a rare subset of cells that function as seeds for new tumor growth in patients with solid tumors. Evidence suggests that CTCs’ identification and characterization could help in predicting the prognosis of breast cancer patients [18,19,20]. CTCs with EMT and stem characteristics have been studied, and it seems that mesenchymal features of CTCs are correlated with a more aggressive disease course and with metastasis formation [17]. Moreover, it was noticed that CTCs from metastatic breast cancer express Programmed Death Ligand 1 (PD-L1), which inhibits the T-cell immune response. This leads to immune system avoidance [21].

In TNBC, cancer cells may hold the characteristics of CSCs both molecularly and functionally; this might explain the poor prognosis of this subtype of breast cancer. Cancer cells in the TNBC subtype express a cluster of differentiation (CD) 44+/CD24− and aldehyde dehydrogenase (ALDH1), this being the signature of breast cancer stem cells and the reason behind tumor initiation, aggressiveness, and therapy resistance [21,22,23]. Due to their small circulating number in blood and the lack of standard strategies for blood pre-processing and CTC isolation and characterization, CTCs’ detection is still challenging [18]. However, despite their scarce presence in whole blood, CTCs’ analysis could contribute to improving the management of cancer patients.

In this article, we will present up-to-date information about the association of CTCs with metastatic triple-negative breast cancer (mTNBC) prognosis and treatment response. We will also discuss the approaches used for CTCs’ detection, highlighting their advantages, disadvantages, and future challenges.

## 2. Methods

We performed a systematic review of studies presenting data about the role of CTCs in TNBC prognosis and treatment response up to February 2021. An initial search was performed in the MEDLINE database searching the following Medical Subject Headings (MeSH) terms: “triple-negative metastatic breast cancer”, “CTC”, and “prognosis” without any filter. A total of 36 articles were found. Then, a second search was conducted in the MEDLINE database using the MeSH terms “breast cancer”, “circulating tumor cells”, and “metastasis”, selecting clinical studies and meta-analyses. Using this search, we identified 151 articles. Finally, an advanced search in the Cochrane library using the terms “triple-negative breast cancer”, “circulating tumor cells”, and “metastasis” in clinical trials led to the identification of 23 articles. A total of 210 articles were found up to February 2021. In January 2022, another search was performed following the same approach as in the initial search. Twenty-four additional articles were found, but only one was eligible for this study. 

The reporting of this systematic review was guided by the standards of the Preferred Reporting Items for Systematic Review and Meta-Analysis (PRISMA) Statement and is currently under assessment at Prospero—registration ID 311495.

Duplicates were removed, and all articles were analyzed based on a number of inclusion and exclusion criteria presented in Table 1.

After a thorough analysis of the 234 articles found, only 24 articles met all the criteria for our review (Figure 1). Our review was conducted using the PRISMA P 2015 checklist [24].

## 3. Results

### 3.1. The Role of CTC in TNBC Prognosis

For a better classification that will eventually lead to a more accurate and personalized treatment, it is important to understand differences in prognosis among different subtypes of metastatic breast cancer. Peeters D.J.E. et al. [19] explored differences in CTC detection and their prognostic significance in terms of OS and PFS between different immunohistochemical subtypes of breast cancer in a retrospective study carried out on 154 MBC patients at baseline, from which 16 were with the triple negative breast cancer (TNBC) subtype. Patients were divided into 5 subgroups based on immunohistochemical subtype, and CTCs were detected in 45.5% of patients. No statistically significant differences were observed between different subtypes of breast cancer and CTC number. A worse prognosis was observed in patients with ≥5 CTCs in 7.5 mL blood in terms of OS and PFS. Patients considered CTC positive had a median OS of 263 months compared to not reached (*p* < 0.001) in those with <5 CTCs, and a median PFS of 9.2 months compared to 17.6 months (*p* < 0.001) in CTC positive and CTC negative patients, respectively. Of lower importance was the association between CTC number and prognosis in the HER2 positive subgroup. The worst prognosis was observed in TNBC patients, particularly in those having ≥5 CTCs, with a median OS of 2.1 months vs. 13.4 months (*p* < 0.009) in CTC negative patients and a median PFS of 2.8 months in CTC positive vs. 25.4 months in CTC negative (*p* < 0.019). On multivariate analysis, from a series of parameters analyzed such as age, CTC status, histological subtype, immunohistochemically subtype, visceral metastasis, bone metastasis, time to metastasis, treatment modality, cancer antigen (CA 15.3), and lactate dehydrogenase (LDH), the only factor associated with shorter OS and PFS was the CTC number ≥5. Shorter PFS was identified in the triple-negative subgroup, and for OS, the presence of visceral disease, early relapse of the disease (<5 years), and triple-negative subtype were unfavorable prognostic factors. Therefore, both a shorter PFS and OS were noticed in metastatic triple-negative breast cancer and in patients having ≥5 CTCs in 7.5 mL blood. At approximately 6 weeks after treatment initiation, half of the patients were evaluated. It was noticed that a decline from ≥5 CTCs at diagnosis to <5 CTCs at follow-up led to an improvement in terms of OS and PFS.

Breast cancer is a heterogeneous group of diseases with a multitude of molecular subtypes, stages, available therapies, and different prognoses, so a lot of classifications were used over the years to simplify the surveillance and therapy selection for these patients. Most of these classifications are complex and create lots of different subgroups; Cristofanilli M. et al. [25] tried to reduce the heterogeneity of metastatic breast cancer classification by performing a large retrospective pooled analysis. Individual patient data of 2436 metastatic breast cancer (MBC) patients, including 358 patients with triple-negative breast cancer (TNBC), from 18 cohorts, were used. The patients were divided into two subgroups based on CTCs, regardless of histology, therapy, or metastasis sites. In this study, patients were identified as being in stage IV aggressive if ≥5 CTCs/7.5 mL blood or in stage IV indolent if the number of CTCs was <5 in 7.5 mL blood. CTC analysis was performed with the CellSearch system and an EpCAM (epithelial cell adhesion molecule) based system, the only Food and Drug Administration (FDA) approved device for CTC analysis. In the first cohort, the collection of samples was performed before the initiation of therapy and included 1944 patients from which 53.1% were considered stage IV indolent and 46.9% stage IV aggressive. In the second cohort of 492 patients, 61.6% were stage IV indolent, and 38.4% were stage IV aggressive. Both in the individual analysis of the cohorts and in the combined one, the ones with stage IV indolent disease had significantly better results in terms of overall survival (OS) compared to those with stage IV aggressive disease (36.3 months versus 16 months) irrespective of the histological subtype, location of the disease, therapy, or metastasis sites. In patients with TNBC, the results were similar with the ones of the combined cohort; those with stage IV indolent disease had a median OS of 23.8 vs. 9.1 months in those with stage IV aggressive disease. In this study, five negative prognostic factors were found on the multivariate analysis including CTC count ≥ 5, triple-negative subtype, the presence of visceral metastasis, grade 3 tumor, and patients who received more than one line of therapy. However, the most significant factor was the CTC count (HR: 2.71, 95% CI (2.35–3.12), *p* < 0.0001).

Lu Y.J. et al. [26] performed a meta-analysis in order to assess the prognostic role of CTCs in metastatic and non-metastatic TNBC patients. Ten studies were eligible for this meta-analysis including 642 TNBC patients. CTC detection methods used among the included studies were CellSearch, EpCAM-based immunomagnetic enrichment/flow cytometry (IE/FC), and reverse transcription polymerase chain reaction (RT-PCR) with CTC cut-offs of either ≥1 or ≥5 CTC/7.5 mL blood. The results showed shorter OS (HR: 2.02, 95% CI = 1.59–2.57, *p* heterogeneity 0.169) and PFS (HR: 2.18, 95% CI = 1.59–2.99, *p* heterogeneity: 0.010) both in metastatic and in non-metastatic TNBC patients that were CTC positive at baseline. This study concludes that the CTC number is an important prognostic tool in TNBC patients regardless of the stage of disease, being statistically significant regarding PFS and with a tendency towards significance regarding OS.

In another study, Dawood et al. [27] assessed the prognostic role of CTC count in 185 MBC patients, in which 61.6% of patients were considered CTC negative and 38.4% CTC positive. Out of 48 patients with the triple-negative subtype, 32 were CTC negative and 16 were CTC positive. In terms of OS, CTC negative patients had a better survival rate (28.3 months) than those who were CTC positive (15 months) (*p* < 0.0001), regardless of the histological or immunohistochemical subtype, metastasis site, and without any differences between the ones that had recurrent or de novo MBC. This study suggested a stratification of breast cancer patients with regards to CTC number.

To evaluate the relationship between CTCs, OS, and progression-free survival (PFS), Wallwiener M. et al. [28] performed a prospective study, in which 468 MBC patients that were about to start a new line of therapy were divided based on immunohistochemical staining into three subgroups as follows: HR+ HER2−: 251 patients, HER2+: 117 patients, and HR−HER2−: 88 patients. About 42% of the patients were found to be positive at baseline. CTC status did not differ significantly among different subtypes of MBC. On the multivariate analysis, several prognostic factors such as line of therapy, the sites of metastasis, and receptor status were found, with positive CTC status being an independent negative prognostic factor in terms of OS and PFS in all the subgroups except the HER2+ subgroup. In HER2+ patients, CTC positivity was a predictive factor only in terms of OS. In the triple-negative subgroup, patients that were CTC positive had shorter PFS than those that were CTC negative (3.05 months compared to 5.83 months, *p* < 0.001); additionally, OS for those who were CTC negative was not reached in contrast to 8.57 months in the CTC positive group (*p* < 0.001).

Similar results were obtained by Munzone E. et al. [29] in a study with 203 patients divided by CTC number into 3 prognostic subgroups: 0 CTC, 1–4 CTC, and ≥5 CTC in 7.5 mL blood, and they were also divided into 5 categories based on tumor characteristics, as follows: luminal A (27 patients), luminal B (105 patients), luminal B- HER2+ (29 patients), HER2+ (24 patients), and TNBC (18 patients). Immunostaining performed for ER, PG, HER2 protein, and proliferation index (Ki-67) was carried out on tissue sections from the primary tumor. CTC quantification was at baseline, performed before a new line of therapy. On multivariate analysis, age, number of metastatic sites, molecular subtype, and bone metastasis were associated with CTC number. The CTC number was an independent prognostic factor for both PFS and OS among all the molecular subtypes, except for TNBC, in which, for OS, CTC number showed a tendency towards significance. This study concluded that CTC quantification plays an important role in stratifying MBC patients based on their prognosis, and this may help to choose a more personalized treatment.

The correlation between CTC number, OS, and time-to-progression (TTP) was analyzed in the study of Mark Jesus M. et al. [30] in which 102 mTNBC patients were included. A new method for CTC analysis called IE/FC was developed, and a comparison between this method and the identification of CTCs using the CellSearch system in a multicenter clinical trial was performed. Blood collection was performed at baseline, before therapy initiation (cetuximab with or without carboplatin), and 7–14 days after. For CTC detection using CellSearch, 7.5 mL of blood were drawn and processed according to manufacturer instructions, while for IE/FC, 10 mL of blood were collected and processed. Eighty-five and 75 patients had both CellSearch and IE/FC CTC quantification at both times. The initial studies on test samples, between IE/FC and the CellSearch system, showed similar data both at baseline (*p* < 0.0001) and at 7–14 days (*p* < 0.0001). The results were expressed as CTCs/mL when using IE/FC and CTCs/7.5 mL when using CellSearch. By using CellSearch, 44% (42 patients) from 95 samples were CTC positive (≥5 CTC/7.5 mL), and 33% (29 patients) out of 89 were CTC positive at 7–14 days. When using IE/FC, at baseline 33% of patients were considered CTC positive, and 34% of patients were CTC positive at 7–14 days. The prognostic impact of CTC number at baseline and at 7–14 days in terms of TTP and OS was analyzed using both methods. In terms of TTP, the CTC number at 7–14 days was of significant importance; meanwhile, CTCs at baseline did not present an impact upon TTP. Regarding OS, CTC positivity both at baseline and at 7–14 days was associated with shorter OS (3 to 6 months) compared to 12 months in CTC negative patients. When the variation of CTC count between the time points (baseline and 7–14 days) was analyzed, it was found that there is a significant association between CTC changes and TTP but only when using the CellSearch system (*p* = 0.03). Regarding OS, a significant association with CTC status using both methods (CellSearch: *p* < 0.0001 and IE/FC *p* ≤ 0.0006) was found; patients with positive CTC count at both times and those who converted from CTC negative into CTC positive had a shorter OS. When the multivariate analysis was performed, the results showed that a positive CTC status at both times using either of the methods was significant in terms of prognosis. The results of this study suggest that CTC numbers at 7–14 days might be a better marker for the risk of progression than CTC counts at baseline, and it showed that the IE/FC method is comparable with the CellSearch system.

In another comparative study for CTC detection, Müller V. et al. [31] evaluated the predictive value of two commercially available techniques used for CTC analysis in 254 MBC patients, including 8 with the TNBC subtype: the CellSearch system and the AdnaTest BreastCancer system. Using the CellSearch system, about 50% of patients were found to be CTC positive. OS was 18.1 months in the CTC positive subgroup in comparison with 27 months in the ones considered to be CTC negative (*p* < 0.001); meanwhile, in relation to PFS, no correlation with CTC positivity was noticed. On multivariate analysis, the number of CTCs was an independent predictor for OS regardless of the disease subtype. In patients with the TNBC subtype, the median OS was 19.7 months in CTC positive patients compared to 26.7 months in CTC negative (*p* = 0.003). Using the AdnaTest BreastCancer system, no correlation was observed between CTC status and OS or PFS, so this study concluded that CTC number is an important prognostic factor in all metastatic breast cancer subtypes using the CellSearch system for their detection, and it also proves the superiority of the CellSearch system in comparison to AdnaTest BreastCancer system.

Riebensahm C. et al. [32] used two methods to identify CTC in 57 patients with breast cancer brain metastasis: CellSearch and an EpCAM-independent method based on Ficoll density centrifugation. About 17.5% of patients had the TNBC subtype. Combining both methods used for CTC analysis, 63.6% of patients were classified as CTC positive. It was noticed that regardless of the method used for CTC identification, a decreased OS of the patients with brain metastasis was associated with an increased CTC number (*p* < 0.05).

Several studies tried to find different markers to ease the prognostic classification of metastatic triple-negative breast cancer patients and to correlate the course of the disease with something measurable. Madic J. et al. [33] conducted a prospective study in 40 mTNBC patients before starting a new line of therapy. They focused on detecting circulating tumor deoxyribonucleic acid (ctDNA) by next-generation sequencing (NGS) based on mutation of the TP53 gene, with high prevalence among TNBC patients, while the CellSearch system was used for CTC detection. ctDNA was not found to have a predictive value for OS or TTP; in contrast, the number of CTCs ≥5/7.5 mL blood at baseline was found to be significant for OS (*p* = 0.04) and almost significant for TTP (*p* = 0.06). Lactate dehydrogenase (LDH) levels and the performing status were also associated with a shorter OS and TTP.

Another study that aimed to evaluate the CTC number and other prognostic parameters in 56 MBC patients, including 10 mTNBC patients, is the one of Helissey C. et al. [34]. Out of the MBC patients included in the trial, 25 (45%) were CTC positive at baseline. It was found that 10 (18%) of the patients had the triple-negative subtype, 5 of them being CTC positive at baseline. CTC count, lymphocyte number, LDH, Barbot’s score (a score that combines Karnofsky performance status, number of metastatic sites, serum albumin level, and LDH), and the prognostic inflammatory and nutritional index (PINI) (a combination of orosomucoid, C reactive protein, albumin, and prealbumin) were assessed before the initiation of a new line of chemotherapy. In a multivariate analysis, the negative prognostic factors at baseline were positive CTC status, triple-negative subtype, poor performance status, and low albumin.

The presence of the CTCs in patients with metastatic breast cancer is a prognostic factor, but during the metastasis process, CTCs’ characteristics might undergo adjustments which may have an impact on the prognosis of the disease. Sara Jansson et al. [35] evaluated the relationship between disease subtype and prognosis of patients with the presence of leukocytes attached to CTCs (WBC-CTC), apoptotic CTCs, or CTC clusters in 52 MBC patients, 4 with the TNBC subtype. Blood draws were performed at baseline, after 1 or 3 months, as well as after 6 months from therapy initiation. CTC clusters were defined as clusters of CTCs containing ≥3 nuclei, apoptotic CTCs as 4’,6-diamidino-2pheylindole (DAPI) stained nuclear morphology, and WBC-CTCs as CTC clusters with ≥1 leukocyte attached. Patients were considered positive if ≥1 CTC cluster or apoptotic CTCs or WBC-CTCs were found. Patients were regularly followed from the baseline until 6 months after the therapy initiation. In all breast cancer subtypes, both OS and PFS were worse in those having ≥5 CTCs/7.5 mL blood. Regarding the presence of WBC or apoptotic CTCs, there was no difference between the disease subtypes. The number of WBC-CTCs was associated with worse OS and PFS, but only at 6 months using univariate analysis; however, after adjustment with some parameters such as age and CTC number, WBC-CTCs seemed to have a favorable impact upon the prognosis. These contradictory results should be analyzed in future studies. The presence of apoptotic CTCs at 1–3 months and 6 months was associated with a worse prognosis in terms of OS and PFS, but no significant association was observed with the baseline number of CTCs. The presence of CTC clusters at baseline and at 1–3 months were found more frequently in patients with TNBC than in the other subgroups, but this difference was not found at 6 months. In terms of prognosis, at 1–3 months, patients with CTC clusters had shorter PFS and OS than those without, and similar results were obtained at 6 months. This study concluded that CTC status is correlated with OS and PFS, and also that apoptotic CTCs and CTC clusters are important prognostic factors, (*p* < 0.001 and *p* = 0.006), so they might be useful for monitoring therapeutic response.

Paoletti C. et al. [36] evaluated the predictive role of CTC numbers, the presence of CTC apoptosis (visual evidence or monoclonal antibody directed against a neo-epitope of the cytokeratin 18 (M30) positive marker for apoptosis in ≥25% of the CTC) and CTC clusters (≥3 nuclei) before starting systemic therapy in 64 mTNBC patients (refer to Table 2). Blood draws were performed at baseline, day 15, and day 29. At baseline, 36.5% of patients were considered CTC positive, from which 26.3% were positive for apoptotic CTCs, and 25% of the patients that had more than 1 CTC presented with CTC clusters. The presence of CTC apoptosis or CTC clusters at baseline was not a significant prognostic factor in terms of PFS; meanwhile, a positive CTC status at baseline was associated with a significantly worse PFS.

The prognostic value of CTC numbers and CTC clusters in terms of PFS and OS was also assessed by Larsson A.M. et al. [37] in 156 patients recently diagnosed with MBC, including 26 with the TNBC subtype before the first line of systemic therapy. CTC identification, enumeration, and CTC cluster (≥2 nuclei) identification were performed using CellSearch, at four different points: baseline, 1, 3, and 6 months after systemic therapy initiation. A total of 52% of patients were considered CTC positive, and 20% of patients had more than 1 cluster at baseline. Out of the 26 patients with the triple-negative subtype, 50% were CTC positive at baseline and 29% had ≥ 1 cluster. In a multivariate analysis, both baseline positive CTC status (*p* < 0.001) and the presence of CTC clusters (*p* < 0.001) were negative prognostic factors in terms of PFS and OS; moreover, it was noticed that the presence of CTC clusters was associated with CTC number, with more than 20 CTC/7.5 mL blood among those having CTC clusters.

The studies presented in this chapter are summarized in Table 2.

### 3.2. The Role of CTC in TNBC Treatment Response

In the study of Helissey C. et al. [34] on 56 MBC patients of which 10 were with the TNBC subtype, a worse prognosis was noticed in terms of PFS in CTCs positive at baseline. Furthermore, a better prognosis was observed in patients with either a decrease of ≥70% from the initial number of CTCs or with fewer than 5 CTCs in 7.5 mL of blood before the second cycle of the third line of chemotherapy. These results suggest that early changes in CTCs under systemic therapy may carry a predictive value for treatment outcomes.

Paoletti C. et al. [36] analyzed the dynamics of CTC number, the presence of CTC clusters, and apoptotic CTCs in 64 mTNBC patients before chemotherapy with nanoparticle albumin-bound paclitaxel with or without adding tigatuzumab. A shorter PFS was noticed in patients who had elevated CTCs at baseline (*p* = 0.005), at day 15 (*p* = 0.0003), and at day 19 (*p* = 0.0002) than those without. Furthermore, impaired PFS was noticed at day 15 (*p* = 0.028) and at day 29 (*p* = 0.009) in patients with CTC clusters; meanwhile the presence of apoptotic CTCs was not a predictive factor for PFS. Patients that were CTC positive at baseline and did not present a clearance of CTCs during therapy had a worse PFS than those whose CTCs decreased both until day 15 (3.6 compared to 1.9 months) (*p* = 0.0003) and day 29 (3.7 compared to 1.9 months) (*p* = 0.0002). Additionally, a better response to therapy was observed in patients whose CTC status cleared by day 15 or 29. This study showed that a decrease in CTC number was prognostic in terms of PFS and showed a response to therapy. Persistence of CTC clusters during treatment was a negative prognostic marker.

Larsson A-M. et al. [37] evaluated the changes in CTC number from baseline to 1, 3, and 6 months from treatment initiation and how these changes influence the outcome of the disease in 156 MBC patients, of which 26 had the TNBC subtype. The odds of disease progression were similar among patients that were consistently CTC negative and those that had a decrease in CTC number from baseline to 1 month. It was noted that patients that were CTC positive at baseline and stayed positive during follow-up had worse PFS at 1 (*p* = 0.002), 3 (*p* = 0.02), and 6 months (*p* < 0.001) than those with a decrease in CTC number. Similar results were obtained for OS; patients who were positive and stayed positive had a worse OS at 1 (*p* < 0.001), 3 (*p* < 0.001), and 6 months (*p* < 0.001). Higher odds of disease progression at the 3-month evaluation were noticed in patients that were CTC positive at 1 month and at 3 months. The evaluation was performed according to RECIST criteria. The presence of CTC clusters was associated with CTC number, as it was noticed that CTC clusters were more frequently detected among patients with more than 20 CTCs in 7.5 mL blood. CTCs’ clusters were associated with decreased PFS and OS. This study concluded that CTC evaluation and the presence of CTC clusters are also important factors for treatment monitoring.

The response to therapy is different among patients with metastatic breast cancer due to a series of factors such as disease heterogeneity and host factors. An early assessment of treatment response could assure an early change in the therapeutic approach in patients with progressive disease in due time. Iwata H. et al. [38] performed a phase III trial to explore the association between CTC number and the effects of different treatments. One hundred forty-eight MBC patients, including 31 with TNBC, were randomized to receive either capecitabine plus docetaxel or docetaxel alone followed by capecitabine when the disease progressed. The blood samples were collected at baseline, before initiating the therapy on day 1 of cycles 2 and 3 and at progression. When the multivariate analysis was performed, the two factors that were associated with a worse prognosis (both PFS and OS) were positive CTC scores at baseline and a triple-negative subtype of the disease. Moreover, a better prognosis in terms of PFS and OS was observed in patients treated with the association of docetaxel and capecitabine. Another result of the multivariate analyses showed that those with a decreased number of CTCs after one cycle of therapy showed a better PFS and OS in comparison with those who had sustained positive CTC status.

Smerage JB. et al. [39] evaluated whether a change in chemotherapy for patients with persistent increased CTCs would be beneficial. At baseline, 319 patients about to start chemotherapy were CTC positive, of which 60 were mTNBC (refer to Table 3). At the first follow-up, 43% had persistently high levels of CTCs. It was shown that they had a worse OS (13 months) and PFS (4.9 months) than those whose CTC number decreased during therapy (OS 23 months, PFS 8.9 months) (*p* < 0.001). Patients with persistently high levels of CTCs were randomly assigned either to continue the initial therapy or to have a change in therapy regimen. The results showed that the change in chemotherapy after one cycle was not beneficial and improved neither OS nor PFS. However, the results of this study confirm the importance of CTC monitorization during therapy in MBC patients, as it was shown that a failure in the decrease in CTCs after therapy initiation was associated with shorter OS and PFS.

Smerage JB. et al. [40] explored not only the role of the CTC number but also the possibility of identifying the CTC expression of two markers: M30 (apoptosis) and anti-apoptotic B-cell lymphoma protein 2 (Bcl-2) and their changes during therapy. The blood draws were performed at baseline, at 1–3 days after the initiation of treatment, and after 3–5 weeks in 83 MBC patients, of which 13 were with the TNBC subtype. Patients were evaluated by computed tomography (CT), magnetic resonance imaging (MRI), and a bone scintigraphy scan before initiating the treatment, and the follow-up was performed on the first day of each cycle. Results showed that an increased number of CTCs at the first follow-up was associated with shorter PFS in comparison with the ones without elevated CTCs (3.4 months compared to 6.4 months) (*p* = 0.0097). Furthermore, in patients with an elevated CTC count, the presence of higher levels of apoptotic CTCs was associated with shorter PFS (0.9 months) in comparison with those that did not present apoptotic CTCs (4.1 months) (*p* = 0.004). When it comes to Bcl2 expression, it was noted as being slightly better prognostically in terms of PFS in Bcl-2 positive CTCs compared to Bcl-2 negative (5.4 months vs. 3.2 months) (*p* = 0.34), but not statistically significant.

Another study which assessed the predictive role of CTC quantification during treatment is the one of Pierga J.-Y. et al. [41]. It compared the predictive significance of some cancer-associated blood markers with the number of CTCs. The CTCs, blood count, LDH, serum calcium, liver enzymes (alanine aminotransferase-ALAT and aspartate aminotransferase-ASAT), gamma-glutamyl transferase (GGT), bilirubin and alkaline phosphatase (ALP), and tumor markers as carcinoembryonic antigen (CEA) and CA 15-3 and Cyfra 21-1 were analyzed at baseline, before cycles 2, 3, and 4 of treatment and when the disease progressed in 265 MBC patients, of which 54 had the TNBC subtype. On multivariate analysis, positive CTC count at baseline was found to be significantly associated with a worse prognosis both in terms of PFS and OS (*p* = 0.03). Moreover, on the multivariate analysis, CTC count before the second cycle of treatment among patients who were positive at baseline was associated with PFS and OS; those having positive CTCs at baseline and maintaining positive CTCs before cycle 2 had a worse prognosis in terms of PFS and OS than those who were positive at baseline and negative before cycle 2 (*p* < 0.0001). When analyzing the CTC count and the serum markers, the results showed that CTC count is an independent factor useful in monitoring treatment management.

In an attempt to clarify the predictive value of CTCs detected by EpCAM isolation techniques, Liu X. et al. [42], in a prospective study, included 75 mTNBC patients who were about to start a new line of therapy. Besides the evaluation of the predictive value of CTCs, the objective was to assess if there is a connection between CTC number, peripheric lymphocyte status, and metastasis. CTC detection was performed using a nanotechnology-based system, Pep@MNP, a system also based on EpCAM isolation similarly to the CellSearch system. Blood draws were performed at baseline. PFS did not differ between CTC positive and CTC negative patients (*p* = 0.118). Phenotypic characterization of the lymphocyte was realized according to the current guidelines. For the natural killer cells (NK)–CTC relationship, a binary regression analysis was used, resulting in two groups: poor prognosis (PFS ≤ 6 months) and favorable prognosis (PFS > 6 months). In first-line therapy patients, baseline CTCs were predictive for PFS (*p* = 0.033) and were positively correlated with lung metastasis (*p* = 0.034) and a number of metastatic sites (*p* = 0.037). Among the whole cohort, baseline CTCs were not predictive of PFS and were not significantly correlated with tumor status or grade, with the number of visceral sites, or histological subtype. As a result of binary logistic regression, PFS was shorter in patients with CTCs > 2/2 mL and NK > 8% (5 months) compared to those with CTCs ≤ 2 and NK >8%, regardless of the therapy line. This study showed that combining CTCs’ enumeration with NK is predictive for PFS in mTNBC patients regardless of the line of therapy (*p* = 0.049), making CTC–NK combined counting a possibly useful prognostic tool in mTNBC patients.

Another study that investigated the role of CTCs in predicting the response to therapy is the one of Liu MC. et al. [43], in which PFS, OS, safety, and overall response rate (ORR) were analyzed in 191 mTNBC patients. The counting of CTCs was performed at baseline and on day 1 of cycles 3 and 5 of chemotherapy. For the evaluation of the CTC dynamic under therapy, patients were classified into subgroup 1 (positive in all evaluation moments), subgroup 2 (positive at baseline and become negative at some point during therapy), and subgroup 3 (negative at baseline). Patients included in group 2 had the best ORR (79.6% vs. 43.8% in group 3 and 29.2% in group 1). Similar results were also obtained regarding PFS (8.5 months in group 2 vs. 5.9 months in group 3 and 4.7 months in group 1) (95% confidence interval (CI), 0.17–0.54). Regarding OS, those in group 1 had the worst prognosis, with an OS median of 9.8 months compared to 16 months in group 3 (95% CI, 0.22–0.73) and 17.8 months in group 2 (95% CI, 0.20–0.62). The results of this study suggested that CTC response to treatment holds a more important prognostic significance than baseline CTCs.

The controversial results about the significance of CTC counting in monitoring the response to therapy among MBC patients, especially those with HER2 positive and TNBC subtypes, led to a prospective study conducted by Jiang Z. F. et al. [44]. The main objectives were to assess the prognostic significance of CTC number and the response to treatment evaluated by the radiographic response. A number of 294 Chinese MBC patients, of which 39 with the TNBC subtype about to start a new line of therapy, were included, 99 controls (women without a history of cancer and any known disease), and 101 women with benign breast disease. Patients were classified as CTC positive at a value of ≥5 CTCs in 7.5 mL blood; the identification of the CTCs was performed using the CellSearch system, and blood was drawn three times: at baseline before receiving a new line of therapy, and at the first and the second follow-up visit. No woman from the control group had >2 CTCs in 7.5 mL blood; meanwhile, 39.1% of patients were CTC positive at baseline. A total of 233 patients had CTC analysis during follow-up. Patients that were CTC positive at baseline had shorter PFS than those that were CTC negative (6.7 compared to 9 months) (*p* < 0.001) and shorter OS (13.2 compared to 24.6 months) (*p* < 0.001). Of 39 patients with metastatic TNBC, 18 were CTC positive at baseline, and they had a shorter PFS and OS in comparison to those that were CTC negative, but not statistically significant. After treatment initiation, 178 patients were CTC negative at the first follow up and they had a better prognosis compared to the 49 CTC positive patients (8.2 months vs. 5.9 months for PFS, *p* = 0.012 and 20.1 months in comparison with 12.4 months for OS, *p* < 0.001). The same trend was noticed at the second follow-up, where 194 patients that were CTC negative had a better prognosis than the 39 CTC positive patients (7.6 months compared to 2 months for PFS and >23.2 months compared to 9.5 months for OS, *p* < 0.001), regardless of subtype. On the multivariate analysis, CTC number at baseline, CTC number at the first follow-up, and CTC number at the second follow-up were independent prognostic factors for both PFS and OS, except for TNBC patients for whom CTC number at baseline was not a significant prognostic factor.

The study of Wallwiener M. [45] et al. evaluated CTCs at baseline, after one cycle of a new line of systemic treatment and CTC kinetics under therapy to assess if these factors can predict response to treatment and the prognosis of the disease in terms of OS and PFS. CTC counting was performed at baseline and after one cycle of systemic therapy in 393 MBC patients, including 57 with the TNBC subtype. About 35% were starting their third or a higher line of therapy. CT or MRI was performed at approximately 3 months after the first cycle of treatment, defining the response into four categories: complete response (CR), partial response (PR), stable disease (SD), and progressive disease (PD) and repeating the evaluation at every 2–3 months until progression was noticed. Based on univariate analysis and reports from previous studies, line of therapy, number of metastatic sites, the site of the metastasis, molecular subtypes, CTC status at baseline, and age were introduced in a multivariate analysis to assess their prognostic significance. When analyzing the relationship between CTC status and survival, significantly shorter PFS was observed for positive CTC baseline status than in patients with negative CTC status at baseline (4.7 compared to 7.8 months) (*p* = 0.001). Similar results were also obtained for OS; those with positive CTCs at baseline had a median of 10.4 months compared to 27.2 months in the negative CTC baseline subgroup (*p* < 0.001). The prognosis was also worse in the CTC positive group after one cycle of therapy, indicating a median PFS of 4.3 months in comparison with 8.5 months in CTC negative (*p* < 0.001), and a median OS of 7.7 months compared to 30.6 months in CTC positive and CTC negative groups, respectively (*p* < 0.001). CTC kinetics was classified as favorable (baseline CTC negative to CTC negative after one cycle of therapy and baseline CTC positive to CTC negative after one cycle of therapy) and unfavorable (baseline CTC negative or positive to positive CTC after one cycle of chemotherapy). The results upon CTC kinetics were not statistically significant for PFS or OS. When the radiological evaluation was performed at 3 months, it was noticed that median OS was 29.9 months for patients with at least SD compared to 13.6 months for patients with PD. From all the factors introduced in the multivariate analysis, positive baseline CTCs, triple-negative subtype, and ≥third line of therapy were significant risk factors for progression and death. For the risk of death only, local, visceral, and bone metastases are also significant risk factors. The main conclusion of this study was that positive baseline CTCs and positive CTC status after one cycle of therapy were significantly associated with both worse PFS and OS.

Liu MC. et al. [46] examined the association between CTCs and radiographic changes in 74 MBC patients, of which 15 were with the triple negative subtype. CTC counting was performed at baseline and 3–4 week intervals. Radiographic examinations were performed at 9–12 week intervals. The results showed that CTC positive patients at 3–4 weeks evaluation had a shorter median PFS (3.1 months) than those that were CTC negative (5.1 months) (*p* = 0.07). Similar results regarding PFS were noticed at 7–9 weeks evaluation after treatment initiation: 2.5 months in CTC positive compared to 6.7 months in CTC negative patients (*p* < 0.001). CTC levels were statistically significantly associated with disease progression, with 7–9 weeks earlier than radiographic changes (*p* < 0.001).

Yan WT. et al. [47] conducted an extensive meta-analysis of 50 studies, including 6721 baseline patients. The therapeutic options included in these studies can be summarized as follows: neoadjuvant therapy, surgery, adjuvant therapy, therapy of metastasis, and combined therapy. They assessed the correlation between the decline in the CTC number and the progression of the disease. The results were difficult to interpret due to the heterogeneity among the included studies, such as different cut-offs for the CTC numbers and different methods for CTC evaluation. Some parameters were adjusted to reduce this heterogeneity. The CTC positivity rate was significantly decreased after treatment, except for surgery. As CTCs are found in peripheral blood, a local treatment as the surgical one cannot eliminate them. This observation suggested the need for different therapeutic strategies other than surgery or postoperative treatment in MBC patients with identifiable CTCs. Another important aspect is that the CTC positive rate decreased after therapy among different molecular subtypes except for the triple-negative one; this observation emphasized the need to discover new treatments for this subtype. The OS was better for patients that had a decreased CTC number after treatment compared to the ones that maintained or increased the CTC during treatment, with a difference of 11.61 months (*p* < 0.00001). Similar results were obtained for PFS, with a mean difference of 5.07 months (*p* < 0.0001) in favor of patients with a decreased CTC number after treatment. A decrease in CTC number was also associated with a lower probability of disease progression.

The studies presented in this chapter are summarized in Table 3.

### 3.3. Approaches Used for CTC Assessment

Obtaining multiple biopsies is difficult in terms of invasiveness for the patient; therefore, CTCs from peripheral blood might be useful in the diagnosis of cancer and cancer recurrence and to monitor treatment efficacy [48]. The majority of systems used for CTC identification have two major steps: enrichment of the CTCs (needed because of their scarcity as there is one CTC at millions of blood cells) and identification of the CTCs [49]. However, we should keep in mind that although evaluation of CTCs can be very helpful, by analyzing only CTCs and not tumor tissue, we cannot assess the tumor microenvironment, which is known to be an important aspect of the disease [50].

Detection of CTCs is a difficult process, and the majority of the currently available methods have their limitations when it comes to maintaining the viability of CTCs; therefore, Mark Jesus M. et al. [30] developed a new method for CTC counting called IE/FC, and they performed a comparison between this method and the identification of CTCs using the CellSearch system in a multicenter clinical trial. The IE/FC has two steps: the first one is an EpCAM-based immunomagnetic enrichment similar to the first step of the CellSearch system. In the second step of the process, when using IE/FC, IE/FC flow cytometric analysis is performed; in this way, cell viability is maintained. In terms of methods comparison, a significant correlation was observed between the two methods studied for CTC counting (CellSearch and IE/FC) both at baseline (*p*< 0.0001) and at 7 to 14 days (*p*< 0.0001). In a multivariate analysis, it resulted that the CellSearch had a better predictive value compared to IE/FC, but the results are comparable.

Another study in search of better methods for CTC detection is the one of Müller V. et al. [31]. They compared the predictive value of two commercially available techniques used for CTC counting: the CellSearch system with the AdnaTest BreastCancer system, which is based on the detection of three tumor-associated transcripts by reverse transcription-polymerase chain reaction (RT- PCR) in metastatic breast cancer patients, especially in those with the HER2+ subtype. The three tumor-associated transcripts used here were MUC1 (mucin 1), HER2, and GA733-2—the messenger ribonucleic acid (mRNA) of the EpCAM. Blood draws were performed before treatment initiation, and samples were processed according to manufacturer guidelines. While using the AdnaTest BreastCancer system, one sample was considered CTC positive if at least one PCR fragment of one of the tumor-associated transcripts (MUC1, HER2, and GA733-2) and a fragment of the internal PCR control gene (gene β-actin) were detected; with the CellSearch system, a sample was considered positive if there were ≥5 CTCs detected in 7.5 mL of blood. Using the AdnaTest BreastCancer system, no correlation was observed between CTC status and OS or PFS. The results of this study suggested the superiority of the CellSearch system in comparison to the AdnaTest BreastCancer system.

Patients with mTNBC have poor prognosis. It is known that MBC patients with brain metastasis possess EpCAM negative CTCs, so Riebensahm C. et al. [32] compared two methods for CTC identification: CellSearch, an EpCAM based technique, and an EpCAM independent technique based on Ficoll density centrifugation. From 57 MBC patients with brain metastasis, 20.5% of patients were CTC positive when the CellSearch system was used; meanwhile, when using the EpCAM independent method, 32% of patients were CTC positive. In mTNBC patients, a superior detection rate was obtained by the independent EpCAM technique. An observation of this study was the fact that in some MBC cases, such as the ones with brain metastasis or the ones with the triple negative subtype, an EpCAM independent method seems to be superior to CellSearch regarding CTC detection, perhaps due to the mesenchymal character of the CTCs in the aforementioned cases.

In an attempt to clarify the predictive value of CTCs detected by EpCAM isolation techniques, Liu X. et al. [42] conducted a prospective study, including patients who were about to start a new line of therapy. The objectives were to evaluate the predictive value of CTCs and to analyze whether there is a connection between CTC number, peripheric lymphocyte status, and metastasis. CTC detection was performed using a nanotechnology-based system, Pep@MNP, a system also based on EpCAM isolation such as the CellSearch system. EpCAM detection is realized via a peptide that is attached through a biotin-avidin interconnection to the iron oxide magnetic nanoparticles; therefore, CTCs expressing EpCAM are identified. This study showed that CTCs-natural killer (NK-CTCs) counting is predictive for PFS in TNBC patients regardless of the line of therapy in comparison with CTC counting alone, perhaps due to a loss of EpCAM expression during systemic therapy, suggesting a possible limitation of EpCAM-isolation-based devices.

Out of the 24 studies, 4 are included in our review on presented CTC identification methods; see Table 4.

## 4. Discussion

Triple-negative breast cancer has a poor prognosis with a predisposition for metastatic progression, and CTCs have a key role in metastasis appearance. In this systematic review of the literature, we set out to investigate the correlation between detection of CTCs in metastatic TNBC patients, prognosis, and response to therapy of these patients. These correlations could be useful in the prediction of tumor aggressivity, choice of therapy at different time points of the disease, and also in monitoring response to treatment. Therefore, we analyzed the role of CTCs in the prognosis of the disease in mTNBC patients in terms of OS and PFS and also their role in the management of these patients.

We presented our thorough analysis of 24 studies that were found to be eligible for our research.

Positive CTC status at baseline seems to be an important independent predictive factor regarding PFS in mTNBC as proven in six of the included studies [19,26,27,28,29,36]. On the other hand, two studies [28,31] did not find CTC status at baseline as being statistically significant for PFS. The contradictory results might be due to the aggressiveness of the disease and also due to the limited number of cases. Moreover, other factors such as the presence of visceral [31] metastasis could influence the prognostic of mTNBC patients beside the CTC number.

Regarding the predictive role of baseline CTC status upon OS only, the results were promising. Positive baseline CTC number was associated with shorter OS in two of the included studies [25,31]. Although that in other two studies [26,29] the results for OS were not statistically significant, baseline CTC status showed a tendency towards significance in mTNBC patients. Another study [32] found that CTC number was an independent prognostic factor for OS in patients with brain metastasis from breast cancer. When the CellSearch system was used, positive baseline CTC status was associated with shorter OS, but by using the AdnaTest BreastCancer system, no connection was noticed [31]. The contrasting results might be due to different systems used for CTC detection and also to the fact that other elements influence the OS such as the presence of visceral metastasis or more than one line of therapy [25].

As positive CTC baseline status alone might not always explain the poor prognosis of the mTNBC, three studies [35,36,37] followed the presence of apoptotic CTCs and CTC clusters beside the CTC count. In two [35,37] out of three studies, the presence of CTC clusters was a negative prognostic factor in terms of PFS and OS. Jansson S. et al. [35] found that the presence of apoptotic CTCs is also related to a worse prognosis of the disease. On the other hand, the results of Paoletti C. et al. [36] showed that the presence of apoptotic CTCs and CTC clusters at baseline was not a significant predictive factor in mTNBC patients.

Aceto N et al. [51,52] studied CTC clusters and noticed that these aggregated cells are rapidly removed from blood flow compared to single cells. Due their large size compared to single CTCs, CTC clusters are trapped in the small capillaries of lungs and distal sites. Moreover, CTC clusters have lower apoptotic rates compared to single CTCs at distant sites and show higher proliferation rates. Their increased cellular viability and rapid clearing at distant sites might explain the increased metastatic potential.

These contradictory data emphasize the need of further investigations of the apoptotic and clustered CTCs in mTNBC patients and also the need to study other CTCs’ markers that could be related to their role in this disease.

Several biomarkers such as CD44+ CD24− and ALDH1+ expression of CTCs are associated with the CSC phenotype and with a more aggressive disease in mTNBC patients [21,22]. Therefore, beside CTC quantification, their biological characterization could provide important predictive information in mTNBC patients.

We found 14 studies in which the role of CTC monitoring upon the treatment response in mTNBC patients is analyzed.

One study [44] showed that a decrease in CTC number at the first and the second follow-up visit during systemic therapy is a significant prognostic factor for both PFS and OS in mTNBC patients. In one study [43], it was shown that the response of the CTC count during treatment seems to be more significant than the baseline CTCs for the prognosis of the disease in mTNBC patients.

However, Yan WT. et al. [47] found that a CTC positive rate decreased after systemic treatment, except for the triple-negative subtype, emphasizing the necessity of finding new management strategies for this subtype.

NK-CTCs [42] as a combined counting tool during therapy held more accurate results than CTC enumeration alone in mTNBC patients regardless of line of therapy.

When it comes to CTC clusters, both Paoletti C. et al. [36] and Larsson A-M. et al. [37] showed that their presence during therapy is associated with a worse prognosis in mTNBC patients.

As CTCs’ detection techniques hold the aforementioned limitations, a combined CTC-ctDNA analysis has the potential to increase the number of patients available for marker monitoring during therapy as both of these markers correlate with treatment response [53].

To sum up, our results underline that positive CTC status at baseline seems to be associated with a worse prognosis in terms of OS and PFS in patients with mTNBC, but their significance when analyzed might be limited due to numerous factors involved in cancer progression [54]. Therefore, CTC quantification combined with other markers, such as the presence of CTC clusters, could improve their predictive role in mTNBC patients. The dynamic of CTC count during the early phases of therapy is an important tool in predicting the response to the treatment. The presence of apoptotic CTCs and CTC clusters could improve the prognostic stratification of these patients even more. However, the identification of CTCs remains difficult due to their rarity, and even though a lot of systems have been developed, there is still a need for better, easier, and faster systems to ease their use in clinical settings and to implement them in the standard of care for mTNBC patients. As CTCs’ phenotype is dynamically changing during mTNBC, the antigen-dependent techniques used for their identification have the limitation of detecting only the CTCs that still express EpCAM in favor of the ones that gained mesenchymal features via EMT. Combined markers, such as NK-CTC, might improve the predictive value in these patients. Moreover, mTNBC response to therapy remains unsatisfactory, so better treatment strategies are much needed in this subtype beside the early assessment of therapeutic response.

## Figures and Tables

**Figure 1 biomedicines-10-00769-f001:**
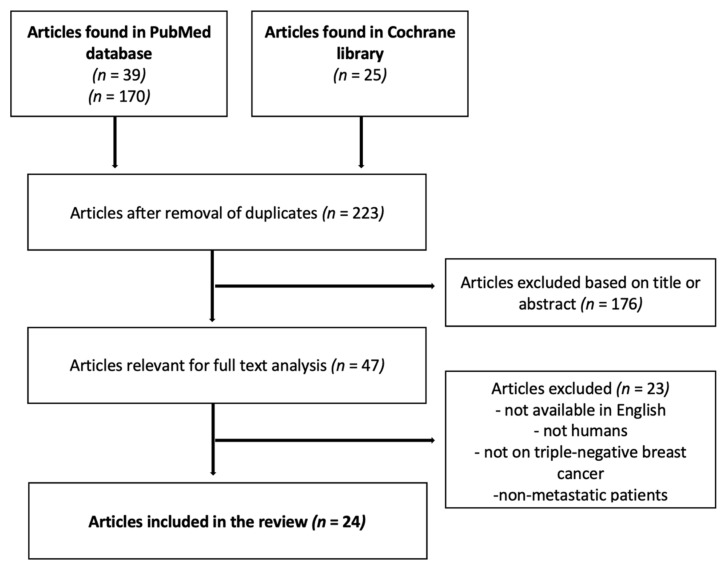
Study selection flow chart. CTC: circulating tumor cell.

**Table 1 biomedicines-10-00769-t001:** Inclusion and exclusion criteria for the analyzed studies.

Inclusion Criteria	Exclusion Criteria
Prospective and retrospective studies	Systematic reviews
Meta-analysis	Non-metastatic breast cancer
Triple-negative breast cancer	Studies that are not in the English language
Metastatic breast cancer	Studies on species other than humans
Studies that are available in the English language	
Species: humans	
The role of CTCs in the prognosis of mTNBC	

Abbreviations: CTCs—circulating tumor cells; mTNBC—metastatic triple-negative breast cancer.

**Table 2 biomedicines-10-00769-t002:** The impact of circulating tumor cells upon the prognosis of the disease in mTNBC ^1^: number of patients, volume of blood, CTC ^2^ threshold, technology used for their characterization, main objective, and main results.

Study	Number of Patients Included	Volume of Blood Analyzed	CTC ^2^ Threshold	CTC Identification System	Main Objective	Main Results	Results Regarding mTNBC ^1^ Patients
1. Peeters D.J.E. et al. [19]	154; 16 mTNBC	7.5 mL	≥5 CTC ^2^/7.5 mL blood	CellSearch	The correlation between CTC number and prognosis among different breast cancer subtypes	CTC positive status is a negative prognostic factor regarding OS ^3^ and PFS ^4^ in MBC ^5^ patients	CTC positive status was associated with shorter OS and PFS
2. Cristofanilli M. et al. [25]	2436; 358 mTNBC	7.5 mL	≥5 CTC/7.5 mL blood	CellSearch	Splitting MBC patients into groups based on their prognosis	Five negative prognostic factors for breast cancer patients: CTC count ≥5, triple-negative subtype, grade 3 tumor, visceral metastasis, and more than one line of therapy	In mTNBC patients, CTC count ≥5 was associated with shorter OS
3. Lu Y.J. et al. [26]	642; Not reported	7.5 mL	≥1, ≥5 CTC/7.5 mL blood	CellSearch IE/FC ^6^ RT/PCR ^7^	Clarifying the prognostic role of CTC in TNBC ^8^ patients	CTC counting is an important prognostic tool in TNBC patients	CTC positive status was statistically significant, associated with shorter PFS and borderline significant for OS
4. Dawood S. et al. [27]	185; 48 mTNBC	7.5 mL	≥5 CTC/7.5 mL blood	CellSearch	Prognostic value of CTC in newly diagnosed MBC patients	Better OS in MBC patients that were CTC negative compared with CTC positive	Better OS in MBC patients that were CTC negative compared with CTC positive
5. Wallwiener M. et al. [28]	468; 88 mTNBC	7.5 mL	≥5 CTC/7.5 mL blood	CellSearch	The correlation between CTC number and prognosis among different breast cancer subtypes	CTC positive status—a negative prognostic factor in terms of OS and PFS	CTC positive status was associated with shorter OS and PFS
6. Munzone E. et al. [29]	203; 18 mTNBC	7.5 mL	0; 1–4; ≥5 CTC/7.5 mL blood	CellSearch	The correlation between CTC number and prognosis among different breast cancer subtypes	CTC positive status—a negative prognostic factor for both PFS and OS in all molecular subtypes, except for TNBC subtype	CTC positive status was associated with shorter PFS, meanwhile regarding OS, the results were borderline significant in mTNBC patients
7. Mark Jesus M. et al. [30]	102; 102 mTNBC	7.5 mL	≥5 CTC/7.5 mL blood	CellSearch	Comparison between two CTC counting methods and the correlation between CTC number and prognosis	CTC number at 7–14 days after therapy initiation was a better marker for prognosis than CTC at baseline; IE/FC is comparable with CellSearch system	CTC number at 7–14 days after therapy initiation was a better marker for prognosis than CTC at baseline; IE/FC is comparable with CellSearch system
		10 mL	≥0.67 CTC/1 mL blood	IE/FC			
8. Müller V. et al. [31]	254; 8 mTNBC	Not mentioned	≥5 CTC/7.5 mL blood	CellSearch	Comparison between two CTC counting methods	CTC positive status was a negative prognostic factor for OS by using the CellSearch system only. For the AdnaTest BreastCancer system, no correlation between CTC and OS or PFS was noticed	CTC positive status was associated with shorter OS when using CellSearch
		Not mentioned	Not mentioned	AdnaTest BreastCancer			
9. Riebensahm C. et al. [32]	57; 10 mTNBC	7.5 mL	≥5 CTC/7.5 mL blood	CellSearch	To assess the genomic alteration involved in the progression of brain metastasis in breast cancer patients	Both methods showed that CTC positive status is associated with a worse OS in patients with brain metastasis of breast cancer	Not specifically mentioned
		7.5 mL	≥1 CTC/7.5 mL blood	An EpCAM ^10^-independent method based on Ficoll density centrifugation			
10. Madic J. et al. [33]	40; 40 mTNBC	5 mL	Not mentioned	NGS ^11^- Ilumina	The prognostic value of CTC compared to ctDNA ^12^	CTC positive status at baseline—negative prognostic factor for OS and borderline significant for TTP	CTC positive status at baseline—negative prognostic factor for OS and borderline significant for TTP
		7.5 mL	≥5 CTC/7.5 mL blood	CellSearch			
11. Helissey C. et al. [34]	56; 10 mTNBC	7.5 mL	≥5 CTC/7.5 mL blood	CellSearch	The prognostic significance of CTC changes in MBC patients	At baseline, negative prognostic factors in terms of OS and PFS—positive CTC status, triple-negative subtype, poor performance status, and low albumin level in MBC patients	Not specifically mentioned
12. Jansson S. et al. [35]	52; 4 mTNBC	7.5 mL	≥5 CTC/7.5 mL blood	CellSearch	The association between CTC count, PFS, and OS	CTC positive status, the presence of apoptotic CTC, and CTC clusters were useful prognostic factors for monitoring the therapeutic response	The presence of CTC clusters at baseline and at 1–3 months of therapy was more frequently found in mTNBC patients
13. Paoletti C. et al. [36]	64; 64 mTNBC	7.5 mL	≥5 CTC/7.5 mL blood	CellSearch	The prognostic role of CTC count, CTC apoptosis, and CTC clusters in MBC	Positive CTC status at baseline—negative prognostic factor in mTNBC patients	Positive CTC status at baseline—shorter PFS in mTNBC patients
14. Larsson A-M. et al. [37]	156; 26 mTNBC	7.5 mL	≥5 CTC/7.5 mL blood	CellSearch	The prognostic impact of CTC number and the presence of CTC clusters in MBC patients	Positive CTC status at baseline and the presence of CTC clusters—negative prognostic factors for OS and PFS in MBC patients	Fifty percent of mTNBC patients were CTC positive at baseline

^1^ mTNBC—metastatic triple-negative breast cancer. ^2^ CTC—circulating tumor cells. ^3^ OS—overall survival. ^4^ PFS—progression-free survival. ^5^ MBC—metastatic breast cancer. ^6^ IE/FC—immunomagnetic enrichment/flow cytometry. ^7^ RT-PCR—reverse transcription polymerase chain reaction. ^8^ TNBC—triple-negative breast cancer. ^9^ TTP—time-to-progression. ^10^ EpCAM—epithelial cellular adhesion molecule. ^11^ NGS—next-generation sequencing.^12^ ctDNA—circulating tumor deoxyribonucleic acid.

**Table 3 biomedicines-10-00769-t003:** The impact of therapy upon circulating tumor cells in mTNBC ^1^: number of patients, volume of blood, CTC ^2^ threshold, technology used for their characterization, and main objective.

Study	Total Number of Patients Included; mTNBC ^1^ Patients	Volume of Blood Analyzed	CTC ^2^ Threshold	CTC Identification System	Main Objective	Main Results	Results Regarding mTNBC ^1^
1. Helissey C. et al. [34]	56; 10 mTNBC ^1^	7.5 mL	≥5 CTC ^2^/7.5 mL blood	CellSearch	CTC dynamic and other palliative prognostic scores	A decrease in CTC number during therapy—better prognosis regarding PFS ^3^	Not specifically mentioned
2. Paoletti C. et al. [36]	64; 64 mTNBC	7.5 mL	≥5 CTC/7.5 mL blood	CellSearch	The prognostic role of CTC count, CTC apoptosis, and CTC clusters in MBC ^4^	A decrease in CTC number during therapy—positive prognostic factor in terms of PFS in mTNBC patients	A decrease in CTC number during therapy—positive prognostic factor in terms of PFS in mTNBC patients
3. Larsson A-M. et al. [37]	156; 26 mTNBC	7.5 mL	≥5 CTC/7.5 mL blood	CellSearch	CTC number and the presence of CTC clusters in the prognostication of MBC patients	A persistent positive CTC status—higher odds of disease progression The presence of CTC clusters—decreased OS ^5^ and PFS	Not specifically mentioned
4. Iwata H. et al. [38]	148; 31 mTNBC	7.5 mL	≥2 CTC/7.5 mL blood	CellSearch	Compare PFS among different therapies	A decrease in CTC number after one cycle of therapy—a better OS and PFS in MBC patients	mTNBC subtype was associated with a worse prognosis in terms of OS and PFS
5. Smerage JB. et al. [39]	595; 134 mTNBC	7.5 mL	≥5 CTC/7.5 mL blood	CellSearch	To evaluate if change in chemotherapy after one cycle in patients with persistent increased CTC would improve the OS	A decrease in CTC number after one cycle of therapy—better OS and PFS in MBC patients	Not specifically mentioned
6. Smerage JB. et al. [40]	83; 13 mTNBC	7.5 mL	≥5 CTC/7.5 mL blood	CellSearch	CTC count, CTC expression of two markers: M30 ^6^ and Bcl-2 ^7^ and the prognosis	Increased number of CTC and the presence of apoptotic CTC—worse prognosis in MBC patients.	Not specifically mentioned
7. Pierga J.-Y. et al. [41]	265; 54 mTNBC	7.5 mL	≥5 CTC/7.5 mL blood	CellSearch	CTC dynamic during therapy and prognosis	Positive CTC status at baseline and sustained CTC positivity during therapy—shorter PFS and OS in MBC patients	Not specifically mentioned
8. Liu X. et al. [42]	75; 75 mTNBC	8 mL	>2 CTC/2 mL blood	Pep@ MNPs assays	The predictive value of CTC count regarding PFS	CTC counting—predictive for PFS only in mTNBC that are undergoing the first line of therapy CTC-NK ^8^ combined counting—predictive for PFS in mTNBC patients regardless of the line of therapy	CTC counting—predictive for PFS only in mTNBC that are undergoing the first line of therapy CTC-NK ^8^ combined counting—predictive for PFS in mTNBC patients regardless of the line of therapy
10. Liu MC. et al. [43]	191; 191 mTNBC	7.5 mL	≥1; ≥2; ≥5 CTC/7.5 mL blood	CellSearch	CTC dynamic under three different chemotherapy regimens	CTC response to therapy holds a more important prognostic significance than baseline CTC status	CTC response to therapy holds a more important prognostic significance than baseline CTC status
11. Jiang Z.F. et al. [44]	294; 39 mTNBC	7.5 mL	≥5 CTC/7.5 mL blood	CellSearch	To evaluate if the ≥5 CTC cut-off is predictive for OS and PFS	MBC patients—CTC number at baseline, at the first follow-up, and the second follow-up were prognostic factors in terms of OS and PFS with the exception of TNBC subtype	In mTNBC patients, CTC number at first follow-up and the second follow-up were significant prognostic factors in terms of OS and PFS
12. Wallwiener M. et al. [45]	393; 57 mTNBC	7.5 mL	≥5 CTC/7.5 mL blood	CellSearch	CTC number and CTC changes during therapy in the prognosis of MBC patients	Baseline CTC status and CTC after 1 cycle of therapy are independent prognostic factors for PFS and OS in MBC patients	mTNBC subtype was an independent prognostic factor for risk of progression and death
13. Liu MC. et al. [46]	74; 15 mTNBC	7.5 mL	≥5 CTC/7.5 mL blood	CellSearch	The correlation between CTC number and radiographic response during therapy in MBC patients	CTC levels were significantly associated with disease progression 7–9 weeks earlier than radiographic changes	Not specifically mentioned
14. Yan WT. et al. [47]	6712; Not mentioned	7.5 mL	≥5 CTC/7.5 mL blood and ≥1/7.5 mL blood	Not reported	The impact of CTC changes during therapy upon prognosis in MBC patients	A persistently high level of CTC during therapy is associated with worse OS and PFS in MBC patients	During therapy, CTC number decreased among different molecular subtypes with the exception of mTNBC subtype

^1^ mTNBC—metastatic triple negative breast cancer. ^2^ CTC—circulating tumor cells. ^3^ PFS—progression-free survival. ^4^ MBC—metastatic breast cancer. ^5^ OS—overall survival. ^6^ M30—monoclonal antibody directed against a neo-epitope of cytokeratin 18. ^7^ Bcl-2—anti-apoptotic B-cell lymphoma protein 2. ^8^ CTC-NK—circulating tumor cell-natural killer.

**Table 4 biomedicines-10-00769-t004:** Studies that analyze CTC ^1^ identification methods.

Study	CTC ^1^ Identification Devices	Results
Mark Jesus M. et al. [30]	Cell Search IE/FC ^2^	CellSearch had a better prognostic value than IE/FC
Muller V. et al. [31]	Cell Search AdnaTest	CellSearch is superior to the AdnaTest
Riebensahm C. et al. [32]	CellSearch An EpCAM ^3^-independent method based on Ficoll density centrifugation	More CTCs were detected by the EpCAM independent method, underlining the possibility that in breast cancer brain metastasis, patients were more EpCAM negative when CTCs are present
Liu X. et al. [42]	Pep@MNPs assays	EpCAM isolation-based devices lose CTC due to the loss of EpCAM expression by the CTC during systemic therapy

^1^ CTC—circulating tumor cell. ^2^ IE/FC—immunomagnetic enrichment/flow cytometry. ^3^ EpCAM—epithelial cellular adhesion molecule.

## Data Availability

Not applicable.

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
