# Peer review of "The Role of Circulating Tumor Cells in the Prognosis of Metastatic Triple-Negative Breast Cancers: A Systematic Review of the Literature"

_biomedicines, 2022, doi:10.3390/biomedicines10040769_

Round 1

Reviewer 1 Report

In this article, authors were proposed interesting interpretation to correlation between metastatic triple-negative breast cancer in blood stream and cancer progression of patients. To performed a systemic analysis to the hypothesis, authors were searched in MEDLINE, Cochrane library, etc. and conducted review using the appropriated procedures.

Basically, authors focused to CTC and prognosis of cancer progression in statistic evidence and specific reports were well summarized in manuscript with arranged tables. Especially, interpretation with anticancer treatment and its prognosis of progression with blood sample were interested and previous literature were appropriated.

-Major comments-

authors should mention about BC stages with references. Although, almost metastatic BC were observed in late stage but early stage of BC also important. Authors have been explained only stage IV.

Author Response

Thank you for your answer and observation!

We have addressed the reviewers observations the following manner:

  • We added BC staging- please see first paragraph from “introduction” section- line 45-55

Please find attached the revised form of our manuscript. We have modified the manuscript according to reviewers’ suggestions. All changes were done with track changes option of Word.

We remain at your disposal if further modifications are required.

Thank you again for considering our work and we hope that this new version is satisfactory!

Reviewer 2 Report

This manuscript by Lisencu et al displays in a very comprehensive way the current challenges and possible clinical utility of CTC detection and – analysis in patients with metastatic triple-negative breast cancer. The systemic search is well-explained and I like the PRISMA checklist. The manuscript is well structured and easy to follow. However, it requires language improvements and some parts should be re-organized. It appears to me that the authors are trying to evaluate the described CTC detection methods. It is very interesting and helpful to receive an update on current CTC detection approaches and the available data. It is certainly of interest to mention advantages, disadvantages and challenges of the detection respective system. However, each system might meet different expectations. I do not believe that there will be a “one fits all” system for future CTC detection. The authors should particularly focus on MTNBC when discussing the CTC detection methods chosen in the included studies relevant for this review manuscript. The discussion is an adequate summary of the systemic review. However, the authors should interpret the discordant results concerning MTNBC. What is the take home message? What novel aspect do the authors conclude from their extensive literature review, except that CTCs are difficult to detect and that better systems are required? What should an improved system aim at? What is needed for MTNBC patients in terms of CTC detection- based on the findings?

I would like to address the following minor issues:

Table 2 and 3.  Why do the authors mention “tissue” ? CTCs were analyzed in blood in all described studies. The colum containing the main results delivers bundled information, but according to the title of the review the authors should add an extra column an briefly summarize the main result concerning MTNBC (keeping in mind the inclusion criteria for their systematic search).

Line 51: spelling: concerning targeted therapies

Line 57: Do all CTCs undergo EMT?

64-66: Do all cancer cells in TNBC have CSC characteristics? Wording should be chosen more carefully.

78 : CTCs instead of CTC. Abbreviations should be used as introduced once.

141:  …with regards to…; use past tense:  This study suggested

171: order of words:  prognostic factor for both, PFS and OS, …

172: What is borderline significant? By trend? Alternatively: …showed a tendency towards significance…? The authors could extend the choice of words such as C”TC quantification” instead of “CTC counting”

202: Abbreviation should be introduced at first use…earlier in the manuscript: line 129

271/272: wording should be more scientific.

284: using univariate analysis

Line 289: structure of sentence is confusing.

Line 296:  Paoletti: What was the used methodology? How much blood was analyzed? Or refer to table 2 at this point.

Line 314: …presence of clusters WAS associated… use past tense.

335: Can personalized treatment be assured? In case, personalized treatment options are available…

334:  the association?

347: Which CTC detection method was used?

354: From the authors report, I understand that therapy change based on persistent CTCs at first follow-up did not show any benefit for the patients. How do the authors draw the conclusion from this study that CTC monitoring during therapy is important? What would be the therapeutic consequence?

363: elevated TC number compared to what? Elevated at first follow up compared to baseline level?

371-381: How many patients and which subtypes were analyzed?

403: past tense and wording; counting of CTCs = CTC quantification or CTC enumeration

407 structure of sentence

423-425 past tense

426 how many were TNBC?

445 wording… included, including…

466: How many patients?

493: wording:  than in negative CTCs… should rather be:  patients with negative CTC status

512: For better flow while reading it would help ig the authors mentionthe number of patients an the applied  CTC detection method for each study? How many TNBC patients were included?

517: CTCs vs. CTC…sentence incomplete…CTC negative PATIENTS.

  1. wording. …of… repeatedly used
  2. wording: …were difficult to interpret due to…

526: wording: … included studies, including…

528 wording. It was noted that sth. was noticed…?? CTC positivity rate was significantly decreased after treatment except for surgery.

529-531 rephrase sentence

534 past tense

Line 553: Tissue biopsy at which state? CNB is a standardized procedure at diagnosis. Tumor tissue can be obtained in most cases. Obtaining multiple biopsies is difficult in terms of invasiveness for the patient.

555: Liquid biopsy MIGHT provide a close-up look. At the time being this is not clinical routine and still controversially discussed. This passage should be carefully re-phrased. Further, the term “Liquid biopsy” does not only refer to CTCs. There are plenty of analytes in various body fluids, which might serve as liquid biopsy, e.g.  ctDNA.

557: Do CTCs in fact hold similar characteristics to the primary tumor? In the introduction section, the authors reported about EMT and that the tumor cells undergo changes causing a variety of CTC populations. This passage should be improved considering the variety of hypothesis on CTC heterogeneity.

559 chemotherapy only?

563 metastasis process-… wording; CTC abbreviation can be used

568: Is the goal to find that ONE best method considering the variety of breast cancer subtypes and therapies and CTC heterogeneity? This should be re-considered.

570 How scarce are they?

583  most frequently used…wording

587 wording CTC that underwent EMT

588 Is cell search that bad? How low is the sensitivity and specificity? Yet, it was approved by the FDA for breast, colon and pancreatic cancer. Are the alternative methods truly less time consuming?

599: What was the sensitivity and specificity of the cellcollector compared to cell search?  Was the cellcollector truly a solution for the low CTC concentration in blood? Wasn’t it rather a novel idea?

613: If the authors claim sth. to be more sensitive or quicker, they should provide facts. How sensitive is this fluorescence-based method and how long does an assay take compared to cellsearch?

661: What is the targeting based on? Is there one ligand for all CTCs?

Discussion:

Line 626- 676: This passage does not belong into the discussion chapter. It deals with technical details concerning various CTC isolation approaches und should be placed into section 3.3.

In the discussion, the authors should focus on the role of CTCs in MTNBC.

667: Patients with TNBC have poor prognosis.

700 past tense

702 what is borderline significant? Instead of repeating facts from previous sections,  they should be interpreted throughout the discussion.

715: …14 studies that followed the relationship?? What do the authors mean?

  1. The authors should discuss what happens during surgery and how CTCs might be influenced during this procedure. What other factors might impact CTC quantification during surgery? There are a couple of theories concerning increased CTC release during surgery and that anesthetic might play a role as well.

723 past tense

738: wording. What is CTC expression?

745 past tense

Author Response

Thank you for your answer and observation!

We have addressed the reviewers observations the following manner:

  • We added the interpretation of the discordant results concerning mTNBC, which are the limitations of CTC detection systems and what a CTC detection system should aim at
  • We removed “tissue” column and we added a column that summarize the main results regarding mTNBC of the included studies- refer to Table 2 and 3
  • We corrected the typing mistakes from initial lines 51,78 now lines 61 and 99
  • We corrected the statement from former line 57 to “a part” of the CTCs now line 68
  • We corrected the statement from former line 64-66 to “TNBC cells may hold the characteristics of the CSCs” now line 85
  • We corrected verb tenses from initial lines 141, 314, 403, 423-425, 534, 700 now lines 203, 453, 616, 509-511, 844, 1005
  • We corrected the errors from initial lines 141, 171, 172, 284 now lines 204, 240, 242, 420
  • We corrected the statement of “borderline significant” initial line 172 to “tendency towards significance” now line 241
  • We corrected the introduction of abbreviation from initial line 129 and 202 now line 189 and 246
  • We corrected the statement “the presence of the CTCs in patients...., CTCs go through a lot of changes” from initial line 271/272 to “the presence of the CTCs in patients..., CTCs characteristics might undergo adjustments which may have an impact on the prognosis of the disease” now line 400/401
  • We corrected the sentence from initial line 289 now line 426
  • We added “refer to Table 2” in initial line 296, now line 436
  • We corrected the statement from “an early assessment of treatment response could assure a more personalised treatment” initial line 335 to “An early assessment of treatment response could assure an early change in the therapeutic approach in patients with progressive disease” now line 530/531
  • We changed from “to explore the relation between CTC number and..” to “to explore the association between CTC number and...” initial line 334 now line 532
  • We added “Refer to Table 3” initial line 347 now line 550
  • We added “as it was showed that a failure in the decrease of CTCs after therapy initiation was associated with shorter OS and PFS.” in initial line 354 now line 557/558 in order to explain why CTC monitoring during therapy is important
  • We added an “increased number of CTCs at the first follow-up” in former line 363 now line 567
  • We changed the sentence from initial line 407 “a better prognosis had patients” to “a better prognostic was observed in patients.” now line 494
  • We added the number of MBC patients and MTNBC patients in the “results” section at each of the included studies for a better flow while reading and the CTC detection method is mentioned in Table 2 and 3
  • We added “patients” in former line 517 now line 766/767 and corrected “CTCs” to “CTC”
  • We corrected the wording error from initial lines 403, 445, 493, 524, 525, 526, 528, 667 now lines 616, 639, 741, 772, 773, 774, 776, 962
  • We rephrased the sentence “other treatment strategies may be more beneficial over surgery” from initial line 529-531 to “As CTCs are found in peripheral blood, a local treatment as the surgical one cannot eliminate them” now line 840- 842
  • We rephrased the sentence “tumor biopsies are sometimes difficult to obtain due to decreased tumor or hard-to-get-to localization” from initial line 553 to “Obtaining multiple biopsies is difficult in terms of invasiveness for the patient” now line 926
  • We rephrased the sentence “moreover, ... can provide a close-up look to tumors’ characteristics” from initial line 555 to “therefore, CTCs from peripheral blood might be useful in the diagnosis of cancer and cancer recurrence and to monitor treatment efficacy” now line 927
  • We removed the phrase “CTC hold similar characteristics to primary tumor cells” from initial line 557 now 927
  • Changed “chemotherapy” to “treatment” former line 559 now line 928
  • We changed the section 3.3 “approaches used for CTC assessment” as suggested by one of the reviewers: we added only the studies included in our research on mTNBC that is the reason why the mistakes and the suggestions for former lines 563, 568, 583, 587, 588, 599, 613 have been removed.
  • Former line 661: The targeting is based on EpCAM and identifies CTCs that are expressing EpCAM as mentioned in lines 967, 970
  • We explained the scarcity of CTCs: “one CTC at millions of blood cells” former line 570 now line 930
  • We changed the place of former lines 626-676 to section 3.3 now lines 926-984 and in “discussion” we focused on interpreting the role of CTC in mTNBC prognosis and treatment
  • We rephrased the statement from initial line 702 now lines 1158
  • We changed the statement from initial line 715 “14 studies that follow the relationship.” to “14 studies in which the role of CTC monitoring upon the treatment response in mTNBC patients is analysed.” now line 1189-1190
  • We added an explanation why CTCs are not removed by surgery (former line 729 now line 777/840. In this section (“Results”) we present the results of the included studies without adding information from the literature
  • As we rewrote the “Discussion” section, error from former lines 723, 738 were removed.
  • “ is associated” former line 745 is now “seems to be associated” line 1209

Please find attached the revised form of our manuscript. We have modified the manuscript according to reviewers’ suggestions. All changes were done with track changes option of Word.

We remain at your disposal if further modifications are required.

Thank you again for considering our work and we hope that this new version is satisfactory!

Reviewer 3 Report

Lisencu et al. present a nice review that addresses an important, up and coming topic, CTCs in breast cancer, here with a focus on TNBC, which is hard to treat and requires novel approaches such as personalized oncology, including better biomarker. The review impresses by the level of detail regarding studies that were included and described. The review is of definitive interest to the community. Several aspects should be addressed though before publication. Please see below.

Comments:

Introduction:

  • The concept of EMT in metastasis is controversial, as e.g., e-cadherin seems to be important in BC. Furthermore, full EMT seems not to be required for metastasis (perhaps an in between state), or even blunt metastatic potential. Please rephrase considering this and cite accordingly
  • Regarding CTC stem phenotype would cite Gkountela et al.
  • CSC phenotype in BC cite Al-Hajj et al.
  • Instead of “counting” CTC would suggest using “detecting” or “enumerating” CTC
  • Would perhaps use mTNBC instead of MTNBC

Methods:

  • Define terms at first use, e.g., MeSH
  • How was eligibility of studies defined (vs. studies were found)?

Results:

  • Please carefully check the use of prognostic vs. predictive
  • Results do not always specifically pertain to mTNBC
  • Perhaps the CTC detection section (3.3.) could be placed before the prognostic and predictive parts of the manuscript?
  • Would not call P = 0.169 a result for OS (page 4, line 131)
  • Please check order of citations (29,30 appear before 25…)
  • Regarding the Mark Jesus M. study (page 5, line 201 and following), how die mTNBC patients fair in this study? Comparing base line with post therapy is also predictive?
  • Please cite Aceto 2014 and 2019 CTC-neutrophil paper (page 7, line 271 and line 294/295 following)
  • The conclusion on page 12, line 355-356 should perhaps also state the current limitation of the predictive value of CTCs.
  • Page 12, line 371 and following, would use predictive instead of prognostic
  • Page 13, line 409-411, why is this considered early changes?
  • Page 13, line 415-416 both spell elevated, please correct
  • Reference 48 is missing
  • References 49 and 50 are not in the bibliography
  • Page 19, line 557, not sure it is adequate to state CTCs are similar to primary tumor cells, perhaps rephrase here
  • Page 19, 563, the role of EMT is not that clear
  • Pag 20, line 608, sentence seems out of context

Discussion

  • This needs some work; I would suggest the following:
  • instead of again repeating part of the results section, start with summary as later done in the discussion
  • Discuss disparity of studies and potential reasons… (example usually baseline CTCs confer worse prognosis, but e.g., Liu et al (40) show never CTCs worse than baseline CTCs positive, Helissey C. et al. (31) show better PFS with baseline pos. CTCs, ….)
  • Include outlook using biological characterization of CTCs as predictive biomarker
  • Include potential combination liquid biopsy (CTCs, ctDNA, etc.)

Author Response

Thank you for your answer and observation!

We have addressed the reviewers observations the following manner:

  • We completed the concept of EMT and what happens to e-cadherin- please see “introduction” lines 69-76
  • We cited Gkonutela and Al- Hajj- please see citations 21-23 for CSC and Aceto (citation 51) for CTC-neutrophil paper
  • We have changed “MTNBC” in “mTNBC”
  • We have changed CTC counting in CTC quantification or detection
  • We have defined MeSH term (former line 80 now line 101)
  • The eligibility of studies was defined based on inclusion and exclusion criteria- please refer to Table 1
  • We have carefully checked and changed “prognostic” to “predictive” where we found it appropriately
  • Results that pertain specifically to mTNBC were added as the last column of Table 2 and Table 3
  • As our review focuses on the role of CTC in the prognosis and treatment response of mTNBC we started the results with addressing these issues and the “approaches used for CTC detection” is a subsection of our study- we corrected this section please see section 3.3
  • We apologise for the order of citations- we made adjustments
  • We corrected former line 131 now line 195-196 by adding “being statistically significant regarding PFS and with a tendency towards significance regarding OS”
  • For the study of Mark Jesus M. similar number of patients had CTC quantification with both methods – we added “Eighty-five and 75 patients had both CellSearch and IE/FC CTC quantification at both times” line 251-252 and “The initial studies on test samples, between IE/FC and the CellSearch system, showed similar data both at baseline (p<0.0001) and at 7-14 days (p<0.0001).” line 254. Yes, the dynamic change between baseline and at 7-14 days of treatment is significant. Please see line 263-267
  • We stated the limitations of the predictive value of CTC – please see Discussion section- line 1176- 1186
  • From initial line 371 now line 576 we changed “prognostic” to “predictive”
  • Former line 409-411 now line 496: early changes in terms of consecutive chemotherapy cycles not in terms of the disease
  • We corrected the errors from initial lines 415-416 now lines 501-502
  • We apologise for the mistakes regarding references, we corrected them
  • We removed the phrase “CTC hold similar characteristics to primary tumor cells” from initial line 557 now 927
  • As we rewrote the 3.3 section- the statement regarding the role of EMT (former line 563) and the sentence that was on line 608 were removed.
  • We made the suggested changes and rewrote the section “Discussion”
  • We discussed the disparity of the studies- please see section “discussion” lines 1003 1162 and further
  • We included an outlook upon the biological features of CTCs – line 1184
  • We mentioned the potential combination of CTCs and ctDNA – please see line 1205-1207

Please find attached the revised form of our manuscript. We have modified the manuscript according to reviewers’ suggestions. All changes were done with track changes option of Word.

We remain at your disposal if further modifications are required.

Thank you again for considering our work and we hope that this new version is satisfactory!

Round 2

Reviewer 2 Report

The authors took all comments into considaration and improved the manuscript accordingly. I recommenda final proov read focussing on a scientific choice of words. Overall, a sound piece of work which is definitely worth the read. 

Reviewer 3 Report

Thank you for addressing my concerns and heeding my suggestions. The manuscript is good shape and can be published.